# Molecular Detection of *Cyclospora cayetanensis* in Two Main Types of Farm Soil Using Real-Time PCR Assays and Method Modification for Commercial Potting Mix

**DOI:** 10.3390/microorganisms11061506

**Published:** 2023-06-06

**Authors:** Joseph Arida, Alicia Shipley, Sonia Almeria

**Affiliations:** 1Office of Applied Research and Safety Assessment (OARSA), Center for Food Safety and Applied Nutrition (CFSAN), Food and Drug Administration, 8301 Muirkirk Road, Laurel, MD 20708, USA; joseph.arida@fda.hhs.gov; 2Joint Institute for Food Safety and Applied Nutrition (JIFSAN), University of Maryland, College Park, MD 20742, USA; 3Department of Biology, University of Rochester, Rochester, NY 14627, USA; ashiple2@ur.rochester.edu

**Keywords:** *Cyclospora cayetanensis*, soil, detection, concentration flotation, silt loam, sandy clay loam, commercial potting mix

## Abstract

*Cyclospora cayetanensis* is a foodborne protozoan parasite that causes outbreaks of diarrheal illness (cyclosporiasis) with clear seasonality worldwide. In the environment, *C. cayetanensis* oocysts are very robust, and contact with contaminated soil may serve as an important vehicle in the transmission of this organism, and it is considered a risk factor for this infection. The present study evaluated a flotation concentration method, previously shown to provide the best detection results when compared with DNA isolation directly from soil samples, in two main types of farm soil, silt loam soil and sandy clay loam, as well as in commercial potting mix samples inoculated with different numbers of *C. cayetanensis* oocysts. The flotation method was able to detect as few as 10 oocysts in 10 g of either type of farm soil without modifications, but needed an extra wash and samples of reduced size for the processing of the commercial potting mix to be able to detect 20 oocysts/5 g. A recently modified real-time PCR method for the detection of *C. cayetanensis* based on a mitochondrial gene target was also evaluated using selected samples of each type of soil. This comparative study confirmed that the concentration of oocysts in soil samples by flotation in high-density sucrose solutions is a sensitive method that can detect low numbers of oocysts in different types of soil.

## 1. Introduction

*Cyclospora cayetanensis* is an important foodborne protozoan parasite responsible for sporadic cases and outbreaks of intestinal disease with clear seasonality worldwide [1,2,3]. In the United States, the parasite has been reported to have caused clinical cases and outbreaks since the mid-1990s. In 2022, there was a total of 1129 laboratory-confirmed domestically acquired cases from 33 states, with 74 of the people affected needing hospitalization [4]. 

Fresh produce, often consumed raw and with little or no washing, has been the main type of commodity implicated in *C. cayetanensis* outbreaks worldwide [3,4]. Fresh produce may become contaminated with oocysts due to poor hygiene practices by food handlers, contaminated soil, and/or contaminated agricultural water [5]. 

Unsporulated *C. cayetanensis* oocysts are excreted by infected humans in feces. Sporulation occurs in the environment. Although there are some studies on the experimental conditions in which *C. cayetanensis* oocysts survive in the laboratory [6,7], unfortunately, there is no information on what triggers oocyst sporulation in field conditions. As is the case with other protozoa, *C. cayetanensis* oocysts are thought to be environmentally resistant; low numbers of oocysts with a heterogenous distribution are expected to be present in soil samples [8,9]. Thus, the detection of *C. cayetanensis* in soil usually requires concentration techniques which enrich the oocyst form of the parasite to enhance detection [9]. Oocysts can contaminate plant crops via different pathways, including the irrigation or spraying of crops with wastewater (black water) and contact with contaminated soil, infected food handlers, or hands that have been in contact with contaminated soil [8]. The detection of oocysts in the soil should not be overlooked due to the risk of human parasitic infections associated with contaminated soil [9]. Several studies, mainly in endemic areas, have implicated contact with soil in the transmission of *C. cayetanensis* [10,11,12,13,14], and soil is increasingly recognized as an important source of *C. cayetanensis* infection in humans. However, there is still a scarcity of available studies and methods for determining the prevalence of *C. cayetanensis* in soil [14,15]. 

Recently, we reported that a concentration method using flotation in saturated sucrose solution provided better *C. cayetanensis* DNA detection rates in soil samples than DNA isolation directly from soil samples using three different commercial kits [8]. The detection of *C. cayetanensis*, as with any other pathogen in soil, poses several problems, including the type of soil analyzed and the presence of inhibitors for molecular detection in soil [16,17,18,19]. Soil and environmental samples with dead biomass may contain substances (humic and fulminic acids, among others), which could inhibit PCR accuracy, even at low concentrations [17]. In addition, differences in parasite recovery among different types of soil have been reported in several studies [9,20,21,22,23,24]. Therefore, it is important to evaluate our previous method using different types of soil. 

Soil composition is of interest because it affects the amount of water stored in the soil that is available to plants. Sandy loam is among the most common farm soil for growing crops [25]. It provides good drainage, allowing for excess water to flow away quickly and easily while nutrients are released to the plants. In addition to sandy loam soil, vegetable and fruit crops will also thrive in silty soils, provided they have adequate drainage. To our knowledge, there have not been any previous comparison studies on the detection of *C. cayetanensis* in different types of soil. In the present study, the previous protocol devised by Shipley et al. [8] was evaluated in two main types of farm soil for vegetable crops: sandy loam soil and silt loam soil collected from two different geographical regions. In addition, the method was also evaluated using a commercial potting mix because this type of soil substrate could be used to grow produce in controlled laboratory conditions using growth chambers, enabling us to investigate the presence of oocysts and the conditions they require for survival in soil. 

## 2. Materials and Methods

### 2.1. Cyclospora cayetanensis Oocysts used and Types of Soil Analyzed 

The oocysts used in the experiments were purified from individual human stool samples from Guatemala and were stored in 2.5% potassium dichromate, as described elsewhere [8]. The study was approved by the institutional review board of the FDA (protocol number 15-039F). The oocysts in the sample used in the present study were found to be unsporulated via microscopy. To enable comparability, the same preparation of oocysts was used for each seeding experiment in the different types of soil samples in the present study. 

Silt loam soil was collected from a farm in Ohio (OH). The composition of this soil was analyzed and found to have a high percentage of silt (65%), a medium percentage of clay (21%), and a low percentage of sand (14%). The sandy clay loam soil was collected from a farm in Georgia (GA), and after analysis it showed a composition of 54% sand, 24% clay, and 23% silt. Similar levels of clay were observed in the two types of soil. Produce grown in the soil collected from OH included, among other crops, berries and leafy greens. The produce grown in the soil collected from GA included mainly leafy greens.

### 2.2. Protocol for Seeding, Flotation Method, and DNA Extraction in Sandy Clay and Silt Loam Soils

The sandy clay and silt loam soil samples were autoclaved prior to the experiments to avoid bacterial and fungal growth [8]. Samples of both types of soil (sandy clay loam and silt loam) weighing 10 g were weighted and aliquoted into 50 mL centrifuge tubes and seeded with different levels of oocysts: 5 samples were seeded with 100 *C. cayetanensis* oocysts, 10 samples were seeded with 20 oocysts, 10 samples were seeded with 10 oocysts, and 5 soil samples were processed unseeded to serve as a negative control, as previously reported [8]. Concentration of the *C. cayetanensis* oocysts was then performed via flotation in high-density sucrose solutions [8]. A total of 30 samples were processed for each type of soil. Briefly, *C. cayetanensis*-seeded and -unseeded control soil samples (10 g) were dispersed by adding 40 mL of deionized water to the 50 mL tube containing the 10 g of soil sample and well mixed manually (1 min). The tubes were then centrifuged at 2000× *g* for 20 min, after which the supernatants were removed. The pellet was mixed with cold sucrose solution (specific gravity (SG): 1.12) until completely mixed. Additional cold sucrose solution was added to fill the tube, and after mixing again, the tubes were centrifuged again at 2000× *g* for 20 min. Supernatant containing the top of the solution (20 mL) was collected into a new 50 mL tube and 30 mL of deionized water was added, followed by centrifugation at 2000× *g* for 20 min. The sediment was retained and washed with deionized water in a 15 mL tube. After centrifugation at 2000× *g* for 20 min, the pellet was transferred to a Fastprep^TM^ (MP Biomedicals, Santa Ana, CA, USA) tube and centrifuged at 14,000× *g* for 4 min, after which the supernatant was removed and the pellet was kept at −20 °C until DNA isolation was carried out. DNA extraction was performed on the two different types of soil using a commercial kit (FastDNA Spin Kit for Soil from MP Bio) in conjunction with a FastPrep-24 Instrument (MP Biomedicals, Santa Ana, CA, USA), following the previously outlined methodology [8]. The original elution volume was 75 µL/sample. 

### 2.3. Protocol Modifications for Processing (Seeding, Flotation Method, and DNA Extraction) in Commercial Potting Mix

Samples from the commercial potting mix were not autoclaved prior to the experiments. Modified approaches were needed to optimize the recovery and detection of the oocysts from the commercial potting mix samples. The top layer of the commercial potting mix containing peat moss, wood chips, and perlite greatly increased the difficulty of the oocyst collection during the flotation step. Therefore, the extra particulate material was removed by carefully pouring off the excess material after centrifugation at 2000× *g* for 20 min in the first wash step of the original protocol, before the sucrose solution flotation step was performed. Once the top layer was removed, the flotation protocol was performed as previously indicated by Shipley et al. [8]. 

Initially, 16 samples (10 g) seeded with 50 oocysts each were analyzed using two different DNA extraction kits: (1) a FastDNA Spin Kit for Soil from MP Bio and (2) a ZymoBIOMICS DNA Miniprep Kit from Zymo Research (Irvine, CA, USA). Similar rates of positive samples and average Cq values were detected using the two DNA kits. High average Cq values and a low percentage of positive samples were detected when the 10 g samples were seeded with 50 *C. cayetanensis* oocysts. Therefore, a smaller sample of commercial potting mix was analyzed (5 g) to compare the levels of detection. Reducing the potting mix sample size to 5 g enabled better detection when the samples were seeded with the same number of oocysts (50 oocysts). To confirm the detection limit in the potting mix, 10 additional 5 g samples were then seeded with 20 oocysts. 

### 2.4. Quantitative Real-Time PCR for Farm Soil and Commercial Potting Mix Samples 

#### 2.4.1. Quantitative Real-Time PCR for Farm Soil and Commercial Potting Mix Samples Using 18S rRNA as Target

After the DNA was extracted from the farm soil and the commercial potting mix, real-time PCR was performed as per Shipley et al. [8]. Briefly, the molecular detection of *C. cayetanensis* was performed via a duplex reaction targeting both the *C. cayetanensis* multicopy 18S ribosomal RNA gene (Cyclo250F *C. cayetanensis* TAGTAACCGAACGGATCGCATT, Cyclo350RN AATGCCACGGTAGGCCAATA, and Cyclo281T *C. cayetanensis* 6-FAM-CCGGCGATAGATCATTCAAGTTTCTGACC) and an exogenous internal amplification control (IAC) (dd-IAC-f IAC CTAACCTTCGTGATGAGCAATCG, dd-IAC-r GATCAGCTACGTGAGGTCCTAC, and dd-IAC-Cy5 IAC Cy5-AGCTAGTCGATGCACTCCAGTCCTCCT) using the QuantiFast Multiplex PCR Kit as the master mix [26]. The qPCR was performed on an Applied Biosystems 7500 Fast Real-Time PCR System (ThermoFisher Scientific, Waltham, MA, USA). A commercially prepared synthetic gBlocks gene fragment (Integrated DNA Technologies, Coralville, CA, USA) (HMgBlock135m) was used as a positive control for the amplification of the *C. cayetanensis* 18S rRNA gene [26,27], and a non-template control (NTC) was included in each run. The amplification protocol consisted of an initial step of 95 °C for 5 min followed by 45 cycles of 95 °C for 30 s and 67 °C for 30 s. Runs were only considered valid if all three replicates of the positive control reactions produced the expected positive result and the NTC was negative. Samples were only considered positive when one or more of the replicates produced a positive result with a cycle threshold (Cq) ≤ 38.0 for the 18S target. According to this method, undetermined reactions would be considered inconclusive if the IAC reaction failed or produced an average Cq value more than three cycles higher than that of the NTC for the same assay, or if the IAC reaction failed completely [26,27]. 

#### 2.4.2. Quantitative Real-Time PCR for Farm Soil and Commercial Potting Mix Samples Using Cox3 (MIt1C qPCR) as Target

A second recently validated (SLV) method for a multi-copy specific target in the mitochondrial genome of *C. cayetanensis* (Mit1C assay) [28] was evaluated in both soils using the lower oocyst levels for comparison with 18S rRNA. The new TaqMan real-time PCR duplex assay targets both the new mitochondrial *C. cayetanensis* gene (Mit1C-f TCTATTTTCACCATTCTTGCTCAC, Mit1C-r TGGACTTACTAGGGTGGAGTCT, and Mit1C-P 6-FAM-AGGAGATAGAATGCTGGTGTATGCACC) and the same exogenous IAC target as described in Section 2.4.1. using PerfeCTa DNA polymerase and multiplex qPCR Toughmix Low ROX reaction cocktail (Quantabio, Beverly, MA, USA) as the master mix [28]. The mitochondrial target in the *Cox3* gene was identified in silico using BLAST searches against *C. cayetanensis* and other genera/species (e.g., *Eimeria* spp. and *Isospora* spp.) in the Apicomplexa phylum. The target was a 205 bp region with a 100% consensus to all reported *C. cayetanensis* sequences in the NCBI database, being species-specific. The amplification protocol consisted of an initial step of 95 °C for 3 min, followed by 40 cycles of 95 °C for 15 s and 61 °C for 1 min, with a cycle threshold (Cq) ≤ 38.0 as a cut off [28]. We tested the new method on selected DNA samples previously analyzed using the 18S rRNA. The samples analyzed were the samples seeded with the lowest levels of oocysts, i.e., silty and sandy clay loam seeded with 10 oocysts from and commercial potting mix seeded with 20 oocysts. Results obtained using 18S rRNA and Mit1C as targets were compared. 

### 2.5. Sequencing of Selected Samples

Selected qPCR positive amplicons from samples of soil spiked with 100 *C. cayetanensis* oocysts (2 samples, 6 positive replicates), 20 *C. cayetanensis* oocysts (2 samples, 6 positive replicates), and 10 *C. cayetanensis* oocysts (1 sample, 2 positive replicates) were combined according to seeding level and sequenced to confirm that *C. cayetanensis* was detected in the presence of the soil background in the samples. The amplicons for sequencing were generated by performing the Mit1C qPCR on these samples without the IAC to avoid the interference of the IAC probe and the primers on the sequencing. The qPCR products were purified using the QIAquick PCR purification Kit (Qiagen, Germantown, MD, USA), following the manufacturer’s instructions. The purified amplicons were sequenced on both strands (Psomagen Inc., Rockville, MD, USA). The sequencing data were then compared with the NCBI nucleotide BLAST suite to confirm the specificity of the amplicons for the detection of *C. cayetanensis* in the soil background. 

### 2.6. Statistical Analysis

Positive detection rates were obtained by calculating the percentage of inoculated samples which gave a positive result in the samples processed via each method and/or experiment. Statistically significant differences in Cq values between two different soil detection levels or between two methods used on the same samples were analyzed using a *t*-test. Statistically significant differences in Cq values between more than 2 soil types were analyzed via one-way analysis of variance and multiple comparisons using Tukey’s multiple comparison test.

Contingency tables for the number of positive samples (percentages) in each type of soil were compared using Fisher’s exact test (two sided). Statistical analyses were performed using GraphPad version 9.1 (GraphPad, San Diego, CA, USA), with a *p* value of ≤0.05 indicating statistical significance. A trend towards statistical significance was considered when the *p* value was ≥0.05 but ≤0.1.

When samples were undetermined in the qPCR reaction, the negative samples were excluded from the calculation of the average Cq values.

## 3. Results

### 3.1. Detection of C. cayetanensis DNA in Silt Loam and Sandy Clay Loam Soil Samples Seeded with Different Numbers of C. cayetanensis Oocysts

#### 3.1.1. Detection of *C. cayetanensis* DNA in Silt Loam and Sandy Clay Loam Soil Samples using 18S rRNA qPCR

The flotation method followed by an 18S rRNA qPCR was able to detect as few as 10 oocysts in 10 g samples of silt loam soil as well as in sandy clay loam samples (Table 1 and Table 2). All of the negative control samples were negative, and no inhibition was observed in any of the processed samples. While the IAC values varied among the samples, they did not diverge from the IAC values of the NTC by 3 Cq values (Table 1). The positivity rates of the silt loam samples were 80% (n = 10), 100% (n = 10), and 100% (n = 5) in the samples seeded with 10 oocysts, 20 oocysts, and 100 oocysts, respectively. In the sandy clay loam soil samples, the respective positivity rates were 20%, 50%, and 80% (Table 2). 

Statistically insignificant differences in the average Cq values for *C. cayetanensis* 18S rRNA were observed between the two types of soil at each of the seeding levels, although a trend towards statistical significance (*p* < 0.1) was observed in the samples seeded with 100 oocysts (35.3 in silt loam versus 36.11 in sandy clay loam, Table 1, *p* = 0.052). 

A significantly higher number of samples seeded with 10 oocysts were detected in the silt loam soil (80% of the 10 samples) compared with the sandy clay loam soil (20% of the 10 samples, *p* = 0.02), and a significantly higher number of samples seeded with 20 oocysts were also detected in the silt loam soil (100% of the 10 samples) compared with the sandy clay loam soil (50% of the 10 samples, *p* = 0.032). In the samples seeded with 100 oocysts, the rate of positive detection was higher in the silt loam (100% of the 5 samples) than in the sandy clay loam (80% of the 5 samples), but the differences were not statistically significant, showing only a trend towards statistical significance (*p* = 0.06).

#### 3.1.2. Detection of *C. cayetanensis* DNA in Silt Loam and Sandy Clay Loam Soil Samples Seeded with 10 *C. cayetanensis* Oocysts (Lowest-Seeded Level) Using Mit1C qPCR

The new Mit1C target qPCR was analyzed using samples of both types of soil seeded with 10 oocysts (limit of detection). For each set of samples seeded (10) in each type of soil, an unseeded soil sample was included in the analysis as a negative control. Among the silt loam samples, a higher number of positive samples seeded with 10 oocysts were detected using the Mit1C qPCR (100% of the 10 samples) than were detected using the 18S rRNA qPCR (80% of the 10 samples) (Table 2), but the difference was not statistically significant (*p* = 0.47). However, statistically significant differences in average Cq values were observed between both targets (37.6 for the 18S rRNA qPCR versus 34.5 for the Mit1C qPCR, *p* = 0.0005). 

Similarly, among the sandy clay loam samples, a higher number of positive samples seeded with 10 oocysts were detected using the Mit1C qPCR (60% of the 10 samples) than were detected using the 18S rRNA (20% of the 10 samples) (Table 2), but difference was not statistically significant (*p* = 0.17). Statistically significant differences were observed in in the average Cq values of both qPCR target methods (36.9 for the 18S versus 32.3 for the Mit1C, *p* = 0.003). 

The average Cq values of the samples of silt loam and sandy clay loam seeded with 10 oocysts analyzed using the Mit1C were statistically significant (34.5 in the silty loam samples versus 32.3 in the sandy clay loam samples, *p* = 0.007). 

### 3.2. Method Modification and Detection Levels of C. cayetanensis DNA in Commercial Potting Mix Samples Seeded with Different Numbers of C. cayetanensis Oocysts

#### 3.2.1. Method Modification and Detection Levels of *C. cayetanensis* DNA in Commercial Potting Mix Using 18S rRNA qPCR

When the flotation method was evaluated using the potting mix, the method needed modifications to achieve optimal detection (see Section 2.3).

Initially, 16 samples (10 g) seeded with 50 oocysts were analyzed using two different DNA extraction kits (MP Bio FastDNA Spin Kit for Soil and ZymoBIOMICS DNA Miniprep Kit) (8 samples each). Statistically insignificant differences were observed in the positivity rates of the samples detected using both of the DNA kits (two of the eight samples were positive using the MP Bio FastDNA Spin Kit for Soil and three of the eight samples were positive using the ZymoBIOMICS DNA Miniprep Kit). Similar average Cq values were also obtained using both DNA extraction kits (an average Cq value of 37.0 was obtained using the MP Bio FastDNA Spin Kit for Soil and an average Cq value of 37.4 was obtained using the ZymoBIOMICS DNA Miniprep Kit, *p* = 0.37). Since high average Cq values and low percentages of positive samples were detected when 10 g samples were seeded with 50 *C. cayetanensis* oocysts, to improve detection, a smaller sample of commercial potting mix was analyzed (5 g). The reduction in the size of the sample enabled better detection (lower Cq values and a higher number of samples detected) in the samples seeded with 50 oocysts (Figure 1a,b). Lower average Cq values were obtained in the 5 g samples of potting mix (average Cq values of 36.6) compared with the average Cq value of 37.2 obtained in the 10 g samples, but the difference was not statistically significant, showing only a trend towards statistical significance (*p* = 0.09). A significantly higher number of positive samples were detected in the 5 g samples of potting mix (100% of the 10 samples at 5 g versus 31.2% of the 16 10 g samples seeded with 50 oocysts, *p* = 0.0007). 

To determine the fractional level of detection and the limit of detection of *C. cayetanensis* in commercial potting mix samples, 10 samples were seeded with 20 *C. cayetanensis* oocysts. The positive detection rate for the samples seeded with 20 oocysts was 20% when using the 18S rRNA qPCR, and therefore 20 oocysts was the detection limit for this type of soil using the modified flotation in sucrose method. 

#### 3.2.2. Method Modification and Detection Levels of *C. cayetanensis* DNA in Commercial Potting Mix Using Mit1C qPCR

Samples seeded with 20 oocysts (the lowest level of detection) were also analyzed using the Mit1C qPCR. An unseeded potting mix sample was included in the analysis as a negative control, and it was found to be undetermined when using the Mit1C. A small increase in the rate of positive detection was observed in the commercial potting mix samples seeded with 20 oocysts using the Mit1C qPCR (30% versus 20% detection rate using the 18S rRNA method, *p* > 0.05), and significantly decreased average Cq values were observed when using the Mit1C method (an average Cq value of 35.2 when using the Mit1C and an average Cq value of 37.2 when using the 18S rRNA qPCR, *p* = 0.015, Table 3). 

### 3.3. Sequencing of qPCR Amplicons from Samples of Soil Seeded with C. cayetanensis Oocysts

Sequencing results from the soil samples seeded with 100 *C. cayetanensis* oocysts showed a partial sequence with a 100% alignment to the reference *C. cayetanensis* mitochondrial genome KP231180 (between 4417bp and 4540bp of the genome). No successful sequencing was achieved using the soil samples seeded with 10 oocysts and 20 oocysts. 

## 4. Discussion

The transmission of *C. cayetanensis* oocysts from contaminated soil is still not well understood. Methods for the detection of the parasite in soil are needed, and such methods need to be shown to work in different types of soil. To our knowledge, there are no previous studies comparing methods for the detection of *C. cayetanensis* in different types of soil. The present study evaluated a previously existing sensitive procedure for the detection of *C. cayetanensis* in large soil samples (5–10 g) [8] using two main types of farm soil, silt loam soil and sandy clay loam soil, as well as a commercial potting mix. Sandy clay loam and silt loam are probably the most common types of farm soil found in the USA in areas where fresh produce linked to *C. cayetanensis* infections are grown. Produce grown in the soil collected in OH included berries and leafy greens, as well as other produce, indicating that vegetable and fruit crops will thrive in silty soils if they have adequate drainage. A commercial potting mix was included in the evaluation of the method since it could be used to grow fresh produce (herbs and berries) in climate-controlled growth chambers or a controlled lab environment and thereby enable us to monitor how detection is affected by time and environmental factors such as temperature, humidity, and/or photoperiod. 

There are still only a few studies concerning the detection of *C. cayetanensis* in soil [14,29,30,31], and to our knowledge, only two of them were carried out using farm soil [27,29], which shows relatively high percentages of positive samples. In the first of these studies, which involved berry farm soils in Mexico, *C. cayetanensis* was detected in 20% (1/5) of soil samples analyzed using nested PCR [31]. Although the soil samples collected at the base of the plants weighed approximately 50 g, only around 250 mg of each soil sample was used for direct DNA extraction and PCR analysis. In the second study, conducted in Apulia, Italy, soil samples were collected from different vegetable/fruit crops planted in succession (cucumber, lettuce, fennel, celery, tomato, melon, endive, and chicory) [29]. Upon the harvesting of each vegetable crop, pooled soil samples (1 kg) were collected from under the water dripper from each plot, and 10 g soil samples were then processed using Percoll–sucrose gradients, commercial genomic DNA isolation kits, and qPCR and melting curve analyses. *Cyclospora* DNA was detected in 11.8% of the 51 soil samples, with the number of *Cyclospora* oocysts estimated (using qPCR) to be from 230 to 661 per 5 μL of genomic DNA. In both studies, the composition of the collected farm soil was not indicated. A recent study involving indigenous people in Venezuela [14] collected soil samples from around huts housing *Cyclospora* cases (2 soil samples from each of the 25 huts) rather than farm soil. The soil was analyzed using UV epifluorescence and phase-contrast microscopy, and 18% (9/50) of the soil specimens were found to be positive for *Cyclospora* sp. [14]. The authors indicated that all the positive soil specimens had a clay texture. Because *Cyclospora* infections predominated in the months of higher rainfall, the authors suggested that mean annual rainfall and the consequent moisture in the soil was the epidemiologically significant factor in the survival of the oocysts and the maintenance of their density in the soil of this semiarid region [14]. In the present study, both of the farm soils analyzed had similar percentages of clay. 

The method/procedure used in the present study was able to detect as few as 10 oocysts/10 g of sample in silt loam soil, as was found in a previous study [6]. Similarly, the silt loam soil analyzed in the present study was collected from the same area and had a similar composition to that used in [6], and the same limit of detection was observed. On the other hand, sandy clay loam with a very different composition also showed the same limit of detection (10 oocysts/10 g of soil). Therefore, the method was consistent in the detection of small numbers of *C. cayetanensis* oocysts in the main types of farm soil, and no inhibition was observed in any of the samples analyzed, regardless of soil type, a finding consistent with the previous study [8]. However, important differences in the detection of the parasite were observed between the two types of farm soil analyzed. Significant differences were not observed in the average Cq values of the two types of farm soil, although a trend towards statistical significance (*p* < 0.1) was observed in the samples seeded with 100 oocysts. The similarity of the average Cq values indicate that, when detected, the parasite was found in similar levels/numbers in the positive silty loam and sandy clay loam samples. However, lower rates of positive samples were observed in the sandy clay loam compared with the silty loam. The sandy clay loam showed lower rates of positive samples at all seeding levels (10 oocysts, 20 oocysts, and 100 oocysts), with statistically significant differences in the lower seeding levels (10 oocysts and 20 oocysts) and a trend toward lower rates at 100 oocysts compared with those of the silt loam samples. Although there were no previous comparison studies on the detection of *C. cayetanensis* in different types of soil, our results confirm previous findings concerning the differences in soil composition as they relate to the detection of protozoan parasites using flotation in soil samples [9,20]. Previous studies on *Toxoplasma gondii* detection in soil reported lower detection/recovery of the parasite in soil with high proportions of sand [9,21,22]. However, in parasites with larger parasitic forms, such as some Taeniid species, *Toxocara canis*, or soil-transmitted helminths eggs, including those of *A. lumbricoides* and *Trichuris trichiura* helminths, higher detection was observed in soil rich in sand [20,23,24]. Therefore, the parasite wall structure and specific gravity of each parasitic form needs to be considered. 

When detection using the flotation method was analyzed in a commercial potting mix, the method needed modifications to enable the elimination of large particles of the potting mix before the proper flotation procedure could be carried out. Potting mix is made up of many less dense components than are found in normal soil, including peat moss, wood chips, and perlite, and these float even in aqueous solutions. These components produced an additional large layer in the supernatant which required removal before the sucrose solution flotation step. In this type of soil substrate, a reduction in the size of the samples, from 10 g to 5 g, significantly increased the detection of *C. cayetanensis* DNA. Therefore, 5 g samples are recommended when processing commercial potting mix. Two commercial kits for DNA extraction after flotation were compared (the MP Bio FastDNA Spin Kit for Soil and the ZymoBIOMICS DNA Miniprep Kit) in this type of soil substrate and were found to have similar detection levels (both in terms of percentage of positive rates and Cq values), but the ZymoBIOMICS DNA Miniprep Kit requires less time to complete the DNA extraction. No inhibition was observed in any of the processed potting mix samples, and after modifications as few as 20 oocysts/5 g of potting mix were detected after the flotation procedure. The ability to detect *C. cayetanensis* oocysts in potting mix will facilitate studies of *C. cayetanensis* detection in growth chambers mimicking environmental field conditions in the laboratory, and this in turn will increase our knowledge of the epidemiology of this important parasite.

Recently, a mitochondrial target for the detection of *C. cayetanensis* using real-time PCR was validated in produce [28]. In the present study, this new target, *Cox3*, was assayed in the sandy clay loam, silt loam, and potting mix samples with the lowest seeding levels and compared with *C. cayetanensis* 18s RNA real-time PCR detection in the same samples to assess sensitivity. Although the rates of positive samples detected were not significantly higher when using the Mit1C qPCR, significantly lower Cq values were observed when using the Mit1C qPCR. In addition, the amplicon of the Mit1C (205 bp) could enable the sequencing of the qPCR fragment (without IAC) to confirm the positivity of the samples. In the present study, the sequencing of positive qPCR samples confirmed the presence of *C. cayetanensis* in the presence of soil background in a sample seeded with 100 oocysts, although we were not able to do so at low seeding levels (10 oocysts and 20 oocysts). Therefore, the use of the second mitochondrial marker could be useful for the detection and confirmation of the presence of *C. cayetanensis* in soil. 

## 5. Conclusions

In conclusion, a method involving flotation in saturated sucrose solution was evaluated using different types of farm soil and showed the same limit of detection in sandy clay loam and silt loam soil, although the sandy clay loam had lower rates of positive detection than the silt loam soil. Method modifications were needed for the commercial potting mix. The modifications evaluated in the present study enabled the detection of small amounts of oocyst DNA in the different types of soil analyzed, and the method could be used to confirm whether soil is a source of *C. cayetanensis* contamination in foods. Since each soil sample can vary in its composition, and since sand content clearly plays a role in the detection of *C. cayetanensis* oocysts, our results should be interpreted with caution. The ability to detect *C. cayetanensis* oocysts in different types of soil will facilitate outbreak investigations and increase our knowledge of the epidemiology of this important parasite. Future investigations could use the present protocol to study the effects of environmental climatic factors in controlled experiments in soil inoculated with *C. cayetanensis*. 

## Figures and Tables

**Figure 1 microorganisms-11-01506-f001:**
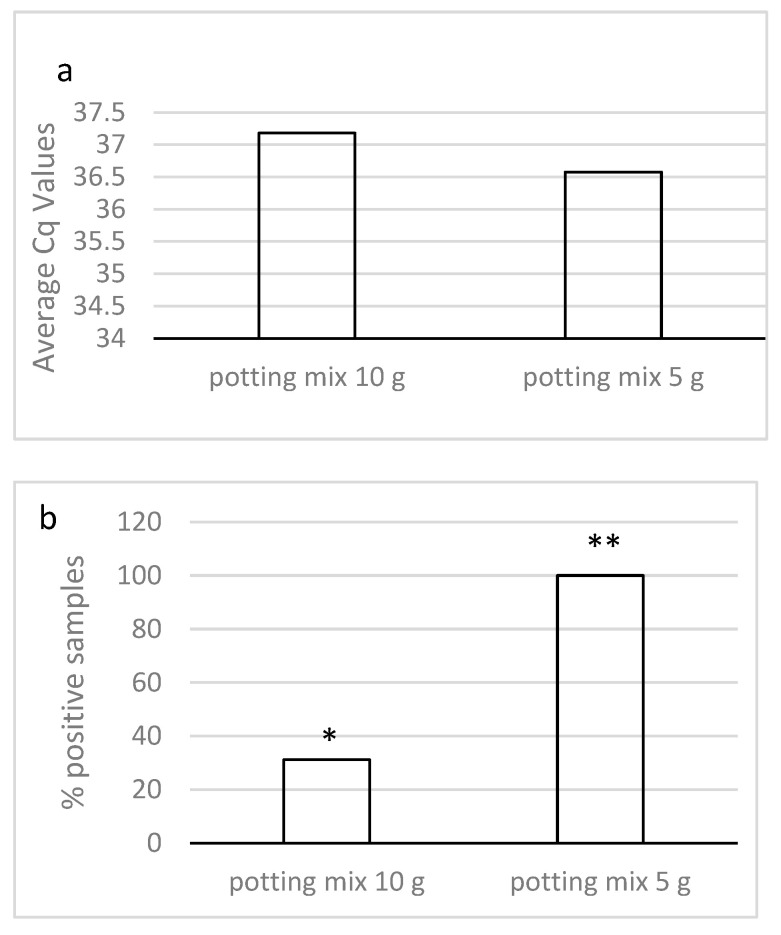
(**a**,**b**) Average Cq values and rate of positivity in 5 g and 10 g samples of commercial potting mix seeded with 50 oocysts. Different asterisks between columns indicates statistically significant differences.

**Table 1 microorganisms-11-01506-t001:** qPCR detection data (number of positive qPCR replicates, individual Cq values) for 18S rRNA *C. cayetanensis* detection in individual samples of sandy clay loam and silt loam soil (10 g), both unseeded and seeded with different numbers of *C. cayetanensis* oocysts (10, 20, and 100) using the flotation in dense sucrose solution method.

No. Oocysts Seeded		Silty Loam (OH)			Sandy Clay Loam (GA)		
	Sample Number	18S rRNA		IAC	18S rRNA		IAC
		No. Positive Replicates	Mean Cq Value	Mean Cq Value	No. Positive Replicates	Mean Cq Value	Mean Cq Value
0	1	0	Und	26.7 ± 0.2	0	Und	25.4 ± 0.1
	2	0	Und	26.5 ± 0.2	0	Und	25.8 ± 0.1
	3	0	Und	26.4 ± 0.2	0	Und	24.8 ± 0.3
	4	0	Und	26.8 ± 0.2	0	Und	25.4 ± 0.1
	5	0	Und	25.8 ± 0.1	0	Und	25.2 ± 0.2
	Average		NA	26.4 ± 0.4		NA	25.3 ± 0.4
10	1	0	Und	26.8 ± 0.3	0	Und	25.1 ± 0.1
	2	1	37.6	26.9 ± 0.3	0	Und	25.4 ± 0.1
	3	3	36.4 ± 0.7	26.8 ± 0.3	0	Und	25.2 ± 0.2
	4	0	Und	26.4 ± 0.1	0	Und	24.8 ± 0.1
	5	1	36.2	26.5 ± 0.1	0	Und	24.8 ± 0.1
	6	1	37.04	26.5 ± 0.1	0	Und	25.4 ± 0.1
	7	3	37.0 ± 1.0	26.4 ± 0.1	0	Und	25.8 ± 0.1
	8	1	37.5	26.1 ± 0.1	0	Und	26.0 ± 0.3
	9	2	36.7 ± 0.7	26.5 ± 0.1	2	36.3 ± 0.7	27.2 ± 2.1
	10	1	36.3	26.6 ± 0.1	2	37.5 ± 0.3	25.2 ± 0.2
	Average		36.8 ± 0.5	26.2 ± 0.3		36.9 ± 0.5	25.5 ± 0.7
20	1	2	37.4 ± 0.4	26.5 ± 0.2	0	Und	27.0 ± 0.7
	2	1	36.9	27.0 ± 0.2	2	37.2 ± 0.5	24.7 ± 0.3
	3	1	36.9	26.7 ± 0.3	0	Und	26.9 ± 0.3
	4	1	37.7	28.2 ± 0.5	1	37.1	26.5 ± 0.1
	5	2	37.3 ± 0.1	26.7 ± 0.3	0	Und	26.8 ± 1.0
	6	2	37.6 ± 0.1	26.5 ± 0.6	1	37.5	26.4 ± 0.0
	7	1	36.8	27.1 ± 0.3	0	Und	26.3 ± 0.3
	8	1	37.0	27.2 ± 0.3	1	36.5	25.9 ± 0.2
	9	2	36.6 ± 0.5	28.0 ± 0.3	0	Und	25.8 ± 0.1
	10	1	36.4	26.5 ± 0.2	2	37.3 ± 0.0	25.8 ± 0.1
	Average		37.2 ± 0.5	27.1 ± 0.1		37.1 ± 0.1	26.5 ± 0.6
100	1	3	34.8 ± 1.0	24.6 ± 0.1	2	35.6 ± 0.6	25.1 ± 0.1
	2	3	35.0 ± 1.5	24.8 ± 0.1	3	36.1 ± 0.8	25.1 ± 0.05
	3	3	36.4 ± 0.1	24.5 ± 0.1	3	36.4 ± 0.2	25.0 ± 0.1
	4	3	35.6 ± 1.2	24.4 ± 0.1	2	36.2 ± 0.6	24.9 ± 0.2
	5	3	35.1 ± 0.6	24.3 ± 0.2	0	Und	24.9 ± 0.2
	Average		35.4 ± 0.6	24.5 ± 0.2		36.1 ± 0.6	25.0 ± 0.1

Und: undetermined, not detected; NA: not applicable.

**Table 2 microorganisms-11-01506-t002:** Average results for detection of *C. cayetanensis* using 18S RNA and Mit1C qPCRs in sandy clay loam and silt loam soil samples (10 g) seeded with different numbers of oocysts of the parasite.

	No. Oocysts Inoculated	No. Samples Analyzed	No. Samples Positive	% Positive Samples	Mean Cq ± Standard Deviation
Silt loam 18S rRNA	0	5	0	0	NA
	10	10	8	80	36.8 ± 0.5
	20	10	10	100	37.2 ± 0.5
	100	5	5	100	35.4 ± 0.6
Silt loam Mit1C	0	1	0	0	NA
	10	10	10	100	34.5 ± 1.8
Sandy clay loam 18S rRNA	0	5	0	0	NA
	10	10	2	20	36.9 ± 0.5
	20	10	5	50	37.1 ± 0.1
	100	5	4	80	36.1 ± 0.6
Sandy clay loam Mit1C	0	1	0	0	NA
	10	10	6	60	33.3 ± 2.6

NA: Not available.

**Table 3 microorganisms-11-01506-t003:** Average results for detection of *C. cayetanensis* using 18S RNA and Mit1C qPCRs in commercial potting mix (5 g) seeded with different numbers of oocysts of the parasite.

Commercial Potting Mix (5 g)	No. Oocysts Inoculated	No. Samples Analyzed	No. Samples Positive	% Positive Samples	Mean Cq ± Standard Deviation
18S rRNA	50	10	10	100	36.4 ± 0.9
	20	10	2	20	37.2 ± 0.3
Mit1C	20	10	3	30	35.2 ± 0.7

## Data Availability

Data are included in the manuscript.

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
