# Peer review of "Molecular Detection of Cyclospora cayetanensis in Two Main Types of Farm Soil Using Real-Time PCR Assays and Method Modification for Commercial Potting Mix"

_microorganisms, 2023, doi:10.3390/microorganisms11061506_

Round 1

Reviewer 1 Report

In this study, the researchers evaluated a flotation method to detect Cyclospora in different types of soil and commercial potting mix samples, and also compared two qPCR methods for detection of Cyclospora in soil. The method was successful in detecting even small numbers of parasite oocysts in farm soil and potting mix samples. The manuscript is well-written and presented, and highly relevant in the field of study. Please consider the following comments for improvement:

Line 42: Produce may also become contaminated with unsporulated oocysts, which then sportulate in the environment. Consider re-wording.

Line 141-142: How exactly was the top layer of extra particulate matter removed? Please provide details.

Recommend using term Cq rather than Ct as per the MIQE guidelines (https://pubmed.ncbi.nlm.nih.gov/19246619/). 

Results:

Did you test the negative/un-spiked soil samples with the Mit1C qPCR? What were the results? If this was completed, please report the results in Table 2. If this was not completed, it is highly recommended to include this data to further support the recommendation for use of the Mit1C target over the 18S target.

Did you do any sequencing of the qPCR products (for the 18S target or Mit1C target) to verify that Cyclospora was detected in the presence of soil background? As this is a novel matrix for both assays (18S and Mit1C), it is important to provide confirmation of presumed positive results wherever possible. 

Line 283-Correct sRNA to rRNA

Line 253,  285- Italicize species name 

Line 358-spell out species names completely when used for the first time. 

Line 386-Since the mit1C data for the unspiked samples was not reported, you cannot comment on whether that assay is more specific than the 18S assay (specificity refers to whether the method can correctly identify a negative sample as negative). Also, although sequencing is mentioned for confirmation, it was not performed or reported.

Figure 1a and b legend should appear under the figure. Lab the Y axes  "Average Cq values" and "% Positive Samples" rather than placing these as a chart title. The x axis labels of 1a are cut off/not fully visible at the bottom. It is not necessary to include the pattern in the bar graph since there is only one data series. 

Author Response

Response to Reviewer 1 Comments

In this study, the researchers evaluated a flotation method to detect Cyclospora in different types of soil and commercial potting mix samples, and also compared two qPCR methods for detection of Cyclospora in soil. The method was successful in detecting even small numbers of parasite oocysts in farm soil and potting mix samples. The manuscript is well-written and presented, and highly relevant in the field of study. Please consider the following comments for improvement.

Response: We thank the reviewer for his/her positive review and helpful comments which we have addressed in full below. Changes are shown in red in the revised manuscript and deletions are shown as strikethrough text.

Line 42: Produce may also become contaminated with unsporulated oocysts, which then sporulate in the environment. Consider re-wording.

Response: The text was reworded mentioning oocysts, and no mention of their stage.  

Line 141-142: How exactly was the top layer of extra particulate matter removed? Please provide details.

Response: We have added the following explanation to how the top layer of extra particulate matter was removed “the presence of the extra particulate material was removed by pouring off the excess material after centrifugation at 2000 × g for 20 min in the first wash step of the original protocol, before the sucrose solution flotation step was performed”

Recommend using term Cq rather than Ct as per the MIQE guidelines (https://pubmed.ncbi.nlm.nih.gov/19246619/). 

Response: The term Ct has been substituted by Cq throughout the manuscript.

Results:

Did you test the negative/un-spiked soil samples with the Mit1C qPCR? What were the results? If this was completed, please report the results in Table 2. If this was not completed, it is highly recommended to include this data to further support the recommendation for use of the Mit1C target over the 18S target.

Response: Thank you very much for your observation. For each set of samples seeded analyzed by Mit1C for each type of soil, including potting mix, a negative control unseeded soil sample was included and analyzed by Mit1C qPCR. This explanation has been included in the text (lines 239-240 and lines 291-292). Results from unseeded soil negative control samples have been added to Table 2.

Did you do any sequencing of the qPCR products (for the 18S target or Mit1C target) to verify that Cyclospora was detected in the presence of soil background? As this is a novel matrix for both assays (18S and Mit1C), it is important to provide confirmation of presumed positive results wherever possible. 

Response: Thank you very much for your observation. We did not try to sequence the 18S rRNA gene target in soil positive samples as the small qPCR fragment precludes efficient sequence amplification. However, the Mit1C qPCR fragment is larger (205 bp) and sequencing was performed in selected positive soil samples from the study (100, 20 and 10 oocysts). A partial sequence was observed in soil seeded with 100 C. cayetanensis oocysts which showed 100% alignment to the C. cayetanensis reference mitochondria genome KP231180 (between 4417bp and 4540bp of the genome). However, in the lower seeded samples, 10 and 20 oocysts, sequencing results were not optimal. This information has now been included in the manuscript.

Line 283-Correct sRNA to rRNA

Response: Corrected

Line 253, 285- Italicize species name 

Response: Italicized the species name.

Line 358-spell out species names completely when used for the first time. 

Response: the species name (Toxoplasma gondii) was spelled out.

Line 386-Since the mit1C data for the unspiked samples was not reported, you cannot comment on whether that assay is more specific than the 18S assay (specificity refers to whether the method can correctly identify a negative sample as negative). Also, although sequencing is mentioned for confirmation, it was not performed or reported.

Response: We modified the text to remove that Mit1C is more specific that the 18S rRNA assay. The focus of the comparison of Mit1C and 18S rRNA in the present study was on sensitivity. We have added the results on un-spiked samples and on the sequencing results to the manuscript, however, we agree that those data are still limited.

Figure 1a and b legend should appear under the figure. Lab the Y axes "Average Cq values" and "% Positive Samples" rather than placing these as a chart title. The x axis labels of 1a are cut off/not fully visible at the bottom. It is not necessary to include the pattern in the bar graph since there is only one data series. 

Response: Following the reviewer’s comments we modified figure 1a and 1b accordingly.

Reviewer 2 Report

The manuscript entitled "Molecular detection of Cyclospora cayetanensis by real-time PCR assays in two main types of farm soil and method modification for commercial potting mix" Title, abstract and overall rationale of work is written satisfactory. There are major concerns, which needs to be addressed before publication.

1) Abstract part is not written well and repeated are there. Author must be re-write the abstract part to clear understanding.

2) Introduction section is written too much lengthy and I recommend author to reduce the size of this content and write concise way.

3) In the material method section: Author must be provide primer details and master mix details with annealing temperature.

4) Results section: The author explained they used two different kind of DNA kits and checked the positivity/sensitivity rate/detection of parasite in the two different soil sample. After PCR analysis of different samples they got CT value of different sample and author compared this CT value to different sample via different methods. However, author do not compared CT values between two sample because the small value change make huge different in the 18s rRNA copy number for example if one sample CT value is 35.2 and other is 35.8 then copy number is totally different both of them and results is significantly different. I recommend author must be calculate this CT value to copy number and then they show the statistical value.

5) Author also need to explain cutoff value of this RT-PCR mean (CT value).

6) Discussion section:  This section author need to improve because author written the results part but they do not discuss properly and I saw the lack of discussion in this manuscript. I recommend author, they should elaborate the discussion part and author need to revise and compare the study with relevant study.

7) Conclusion section must be write separate section. Moreover, in this conclusion section author must be write the limitation of this study and significance. Moreover, author also need to write future prospective of this study.

8) Some references are too long and author need to revise for example reference no 13, 20 and other. I suggest author to revise if other latest manuscript is available in the same information.

9) In the title section; author should be write specie name in italic.  

English is okay.

Author Response

Response to Reviewer 2 Comments

The manuscript entitled "Molecular detection of Cyclospora cayetanensis by real-time PCR assays in two main types of farm soil and method modification for commercial potting mix" Title, abstract and overall rationale of work is written satisfactory.

Response: We thank the reviewer for his/her positive review and helpful comments which we have addressed in full below. Changes are shown in red in the revised manuscript and deletions are shown as strikethrough text.

There are major concerns, which needs to be addressed before publication.

1) Abstract part is not written well and repeated are there. Author must be re-write the abstract part to clear understanding.

Response: The abstract was slightly modified to avoid any repetition. We will be happy to further modify the abstract if the reviewer indicates which parts need clearer understanding.

2) Introduction section is written too much lengthy and I recommend author to reduce the size of this content and write concise way.

Response: The introduction section has been reduced in length. Deleted text is shown as strikethrough text in the manuscript.

3) In the material method section: Author must be provide primer details and master mix details with annealing temperature.

Response: Details of primers, probes, master mixes and annealing temperature have been added in the material and methods section for both methods.

4) Results section: The author explained they used two different kind of DNA kits and checked the positivity/sensitivity rate/detection of parasite in the two different soil sample. After PCR analysis of different samples they got CT value of different sample and author compared this CT value to different sample via different methods. However, author do not compared CT values between two sample because the small value change make huge different in the 18s rRNA copy number for example if one sample CT value is 35.2 and other is 35.8 then copy number is totally different both of them and results is significantly different. I recommend author must be calculate this CT value to copy number and then they show the statistical value.

Response: Thank you for your suggestion. We performed statistics of the 18s rRNA copy numbers at the different seeding levels between the two types of soil and results showed the same statistical significance/trends as those shown by Ct values. Copy numbers of C. cayetanensis 18S rDNA or Mit1C detected with the qPCR assay at each sample are calculated by extrapolation of their Ct values on a standard curve, and therefore CT values and gene copy numbers are dependent variables. We have not included those results in the text of the manuscript to avoid duplication.

5) Author also need to explain cutoff value of this RT-PCR mean (CT value).

Response: The cut-off information for Mit1C qPCR has been added in the Material and Methods section.

6) Discussion section:  This section author need to improve because author written the results part but they do not discuss properly and I saw the lack of discussion in this manuscript. I recommend author, they should elaborate the discussion part and author need to revise and compare the study with relevant study.

Response: The Discussion section compared and discussed the results found in the present study based on a comprehensive review of the most relevant studies in soil published to date on C. cayetanensis as well as in other parasites in relevant studies on the subject. If the reviewer would like us to add some additional relevant studies, we would be happy to do that if he/she could indicate which ones.

7) Conclusion section must be write separate section. Moreover, in this conclusion section author must be write the limitation of this study and significance. Moreover, author also need to write future prospective of this study.

Response: The conclusion has been separated in a different section of the text and the limitations of the study as well as the relevance/significance and the future prospective of the study have been included in this section.

8) Some references are too long and author need to revise for example reference no 13, 20 and other. I suggest author to revise if other latest manuscript is available in the same information.

Response. We believe the reviewer is referring to the older references in the present study. There are only three old references from the 90s, and after careful consideration we believe that they are of importance for the present study. Reference 6 (Smith 1997) is one of the few studies performed to date showing which conditions are needed for sporulation of C. cayetanensis. Reference 13 (Koumans et al 1998) showed for the first-time soil contact as a risk factor in a C. cayetanensis outbreak, and reference 20 (Nunes et al 1994) refers to flotation studies for parasite detection which is the method used in the present study. The rest of the references included in the manuscript are recent, from 2000 onwards. If the reviewer would like us to add some additional references, we would be happy to do that if he/she could indicate which ones.

9) In the title section; author should be write specie name in italic.  

Response: The species names have been indicated in italics in the tittle sections.

Round 2

Reviewer 1 Report

All suggested revisions have been carefully addressed. 

Author Response

 We thank the reviewer for his/her careful review and helpful comments.

Reviewer 2 Report

The authors have addressed all the concerns raised in the previous version of the manuscript and the quality has much improved after incorporating required modifications. The author change the Ct to Cq but in the figure they do not change. Kindly change this to figure 1 also and please write the legend below the figure. Therefore, the manuscript may be considered for publication in this Journal.

Author Response

The authors have addressed all the concerns raised in the previous version of the manuscript and the quality has much improved after incorporating required modifications. The author change the Ct to Cq but in the figure they do not change. Kindly change this to figure 1 also and please write the legend below the figure. Therefore, the manuscript may be considered for publication in this Journal.

Response: We thank the reviewer for his/her careful review and helpful comments which we have addressed in full below. The Ct to Cq has been changed in Figure 1 and the legend is now written below the figure.